# Prevalence of sexual dimorphism in mammalian phenotypic traits

Natasha A. Karp[1,2], Jeremy Mason[3], Arthur L. Beaudet[4], Yoav Benjamini[5], Lynette Bower[6], Robert E. Braun[7], Steve D.M. Brown[8], Elissa J. Chesler[7], Mary E. Dickinson[9], Ann M. Flenniken[10], Helmut Fuchs[11], Martin Hrabe de Angelis[11,12,13], Xiang Gao[14], Shiying Guo[14], Simon Greenaway[8], Ruth Heller[5], Yann Herault[15,16,17,18,19], Monica J. Justice[20], Natalja Kurbatova[5], Christopher J. Lelliott[21], K.C. Kent Lloyd[6], Ann-Marie Mallon[8], Judith E. Mank[22], Hiroshi Masuya[23], Colin McKerlie[10,24], Terrence F. Meehan[3], Richard F. Mott[25], Stephen A. Murray[7], Helen Parkinson[3], Ramiro Ramirez-Solis[21], Luis Santos[8], John R. Seavitt[4], Damian Smedley[26], Tania Sorg[15,16,17,18,19], Anneliese O. Speak[21], Karen P. Steel[21,27], Karen L. Svenson[7], The International Mouse Phenotyping Consortium[†], Shigeharu Wakana[23], David West[28], Sara Wells[8], Henrik Westerberg[8], Shay Yaacoby[5], Jacqueline K. White[7,21]

The role of sex in biomedical studies has often been overlooked, despite evidence of sexually dimorphic effects in some biological studies. Here, we used high-throughput phenotype data from 14,250 wildtype and 40,192 mutant mice (representing 2,186 knockout lines), analysed for up to 234 traits, and found a large proportion of mammalian traits both in wildtype and mutants are influenced by sex. This result has implications for interpreting disease phenotypes in animal models and humans.

[1] Mouse Informatics Group, The Wellcome Trust Sanger Institute, Hinxton, Cambridge CB10 1SA, UK. [2] Quantitative Biology, AstraZeneca, Unit 310, Cambridge Science Park, Cambridge CB4 0WG, UK. [3] European Bioinformatics Institute (EMBL-EBI), European Molecular Biology Laboratory, Wellcome Trust Genome Campus, Hinxton, Cambridge CB10 1SD, UK. [4] Human and Molecular Genetics, Baylor College of Medicine, 1 Baylor Plaza, Houston, Texas 77030, USA. [5] Department of Statistics and O.R. School of Mathematical Sciences, Tel Aviv University, Tel Aviv 69978, Israel. [6] Mouse Biology Program, University of California, 2795 Second Street, Suite 400, Davis, California 95618, USA. [7] The Jackson Laboratory, 600 Main Street, Bar Harbor, Maine 04609, USA. [8] MRC Harwell Institute, Harwell Campus, Harwell OX11 0RD, UK. [9] Molecular Physiology and Biophysics, Baylor College of Medicine, 1 Baylor Plaza, Houston, Texas 77030, USA. [10] The Centre for Phenogenomics, 25 Orde Street, Toronto, Ontario, Canada M5T 3H7. [11] German Mouse Clinic, Institute of Experimental Genetics, Helmholtz Zentrum München, Ingolstädter Landstraße 1, Neuherberg 85764, Germany. [12] School of Life Science Weihenstephan, Technische Universität München, Alte Akademie 8, Freising 85354, Germany. [13] German Center for Diabetes Research (DZD), Ingostädter Landstr. 1, Neuherberg 85764, Germany. [14] Model Animal Research Center, Nanjing University, 12 Xuefu Road, Pukou, Nanjing, Jiangsu 210061, China. [15] CELPHEDIA, PHENOMIN, Institut Clinique de la Souris, 1 Rue Laurent Fries, Illkirch 67404, France. [16] Institut de Génétique et de Biologie Moléculaire et Cellulaire, 1 Rue Laurent Fries, Illkirch 67404, France. [17] Centre National de la Recherche Scientifique, UMR7104, 1 rue Laurent Fries, Illkirch 67404, France. [18] Institut National de la Santé et de la Recherche Médicale, U964, 1 rue Laurent Fries, Illkirch 67404, France. [19] Université de Strasbourg, 1 rue Laurent Fries, Illkirch 67404, France. [20] The Hospital for Sick Children, 555 University Avenue, Toronto, Ontario, Canada M5G 1X8. [21] Mouse Genetics Project, The Wellcome Trust Sanger Institute, Hinxton, Cambridge CB10 1SA, UK. [22] Department of Genetics, Evolution & Environment, University College London, Gower Street, London WC1E 6BT, UK. [23] BioResource Center, RIKEN, 3-1-1 Koyadai, Tsukuba, Ibaraki 305-0074, Japan. [24] The Hospital for Sick Children, 686 Bay Street, Toronto, Ontario, Canada M5G 0A4. [25] Genetics Institute, University College London, Gower Street, London WC1E 6BT, UK. [26] Clinical Pharmacology, Queen Mary University of London, Gower Street, London WC1E 6BT, UK. [27] Wolfson Centre for Age-Related Diseases, King's College London, Wolfson Wing, Hodgkin Building, Guys Campus, London SE1 1UL, UK. [28] Children's Hospital Oakland Research Institute, 5700 Martin Luther King Jr Way, Oakland, California 94609, USA. Correspondence and requests for materials should be addressed to N.A.K. (email: natasha.karp@astrazeneca.com).
[†]A full list of consortium members appears at the end of the paper.

A systematic review of animal research studies identified a vast over-representation of experiments that exclusively evaluated males. Where two sexes were included, two-thirds of the time the results were not analysed by sex[1,2]. Furthermore, sex is often not adequately reported, despite the majority of common human diseases exhibiting some sex differences in prevalence, course and severity[3]. Fundamental differences exist between males and females that may influence the interpretation of traits and disease phenotypes[4,5] and their treatment[6]. Some, however, have argued that considering both sexes lead to a waste of resources and underpowered experiments[7], while others have questioned the value of preclinical research into sex differences[8].

Here we quantify how often sex influences phenotype within a data set by analysing data from 14,250 wildtype animals and 40,192 mutant mice, from 2,186 single gene knockout lines, produced by the International Mouse Phenotyping Consortium (IMPC)[9]. The phenotyping performed by the IMPC explores a range of vertebrate biology, and aims to collect data from seven males and seven females from each mutant line with data from strain-matched controls accumulated over time. Data are collected at 10 phenotyping centres, providing a unique opportunity to explore the role of sex on a phenotype within an experimental data set, and the role of sex on a treatment effect, where the treatment is a gene disruption event, analogous to a Mendelian genetic disease. Our findings show that regardless of research field or biological system, consideration of sex is important in the design and analysis of animal studies. All data are freely available at mousephenotype.org.

## Results

**Sex as a biological variable within an experiment.** We first assessed the contribution of sex using linear modelling to determine how often sex contributed to the variation in the phenotype in wildtype mice (control data) for an individual data set (a phenotypic test/trait at an individual phenotyping centre) (Supplementary Fig. 1a,b). Phenotypes were classified as either continuous, such as creatine kinase levels, or categorical, such as vibrissae shape. Because body size is dimorphic between male and female mice, and many continuous traits correlate with body weight, we included weight as a covariate in our analysis for continuous traits. Using this approach, our analysis revealed that 9.9% of data sets from categorical traits (54/545 data sets) were significantly influenced by sex at a 5% false discovery rate (FDR) (Fig. 1a). Many of these cases included phenotypes that would not *a priori* be assumed to be sexually dimorphic (SD). For example, abnormal corneal opacity occurred at a higher rate in female wildtype mice at most phenotyping centres. Looking at the SD rate by institute, we find that within categorical data the rate was relatively consistent (average percentage of traits which were SD 8.9% (s.d. = 5.9)) (Fig. 1c).

For continuous traits, a far higher proportion of data sets (56.6%, 511/903) exhibited sexual dimorphism at a 5% FDR (Fig. 1b). As expected, this proportion was higher when the absolute phenotypic differences were considered without taking body weight into account (73.3%, 662/903 data sets, Fig. 2a). With the continuous data set, the inter-institute SD rate was more variable (average percentage of traits which were SD 44% (s.d. = 14)) (Fig. 1d). Variation in sensitivity is to be expected, arising from the observation that variance for a trait depends on the institute[10] and the size of the control data set (Fig. 2b,d). Regardless of biological area studied, sex was found to have a role (Figs 2c and 3a,b) and where calls could be compared across institutes the effect of sex was in general reproducible, with only 8.7% of variables having opposing effects across the phenotyping centres (Fig. 3c,d). Variation in husbandry, diet and other

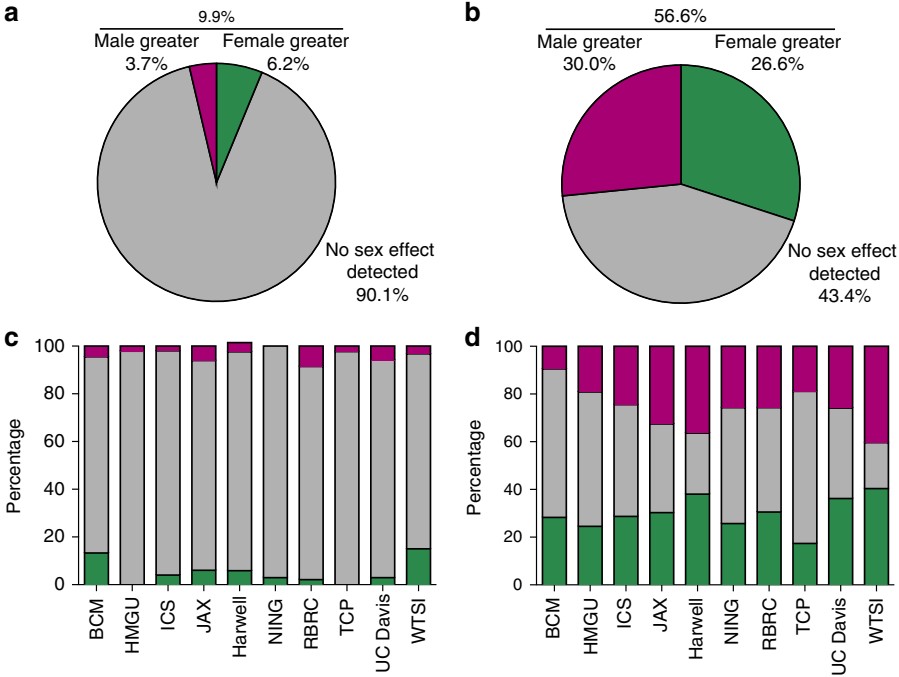

**Figure 1 | Sex as a biological variable in control data.** The role of sex in explaining variation in phenotypes of wildtype mice as assessed using data from the IMPC. (**a,b**) The proportion of experiments where sex had a significant role in wildtype phenotype. (**a**) Categorical data sets (n = 545). (**b**) Continuous data sets (n = 903). (**c,d**) The distribution of classifications when analysed by institute (c: categorical data sets, d: continuous data sets). BCM: Baylor College of Medicine, HMGU: Helmholtz Zentrum Munich, ICS: Institut Clinique de la Souris, JAX: The Jackson Laboratory, Harwell: Medical Research Council Harwell, NING: Nanjing University, RBRC: RIKEN BioResource Centre, TCP: The Centre for Phenogenomics, UC Davis: University of California, Davis, and WTSI: Wellcome Trust Sanger Institute.

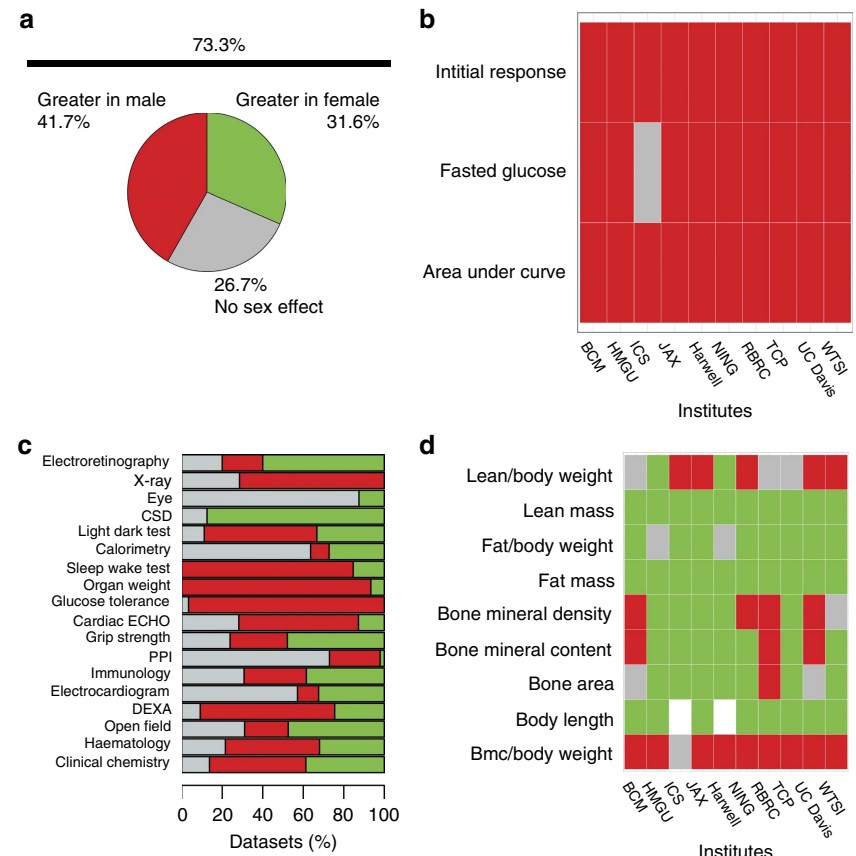

**Figure 2 | Sex as a biological variable in wildtype phenotypic continuous data when exploring absolute difference in phenotypes.** Exploration of how often sex was significant at explaining variation at a 5% FDR in an individual experiment using IMPC wildtype data for continuous traits. The analysis assessed the role of sex in the trait of interest, at a centre level, as an absolute phenotype since weight was not included as a covariate. For all sections, green indicates the phenotype was greater in the female, magenta indicates the trait was greater in the males, white indicates missing data, and grey indicates there was no significant sex effect. (**a**) Pie chart showing the proportion of data sets where sex was a significant source of variation ($n = 903$). (**b**) Comparison of the reproducibility of the sex differences in the traits monitored within the intra-peritoneal glucose tolerance test across ten phenotyping centres. (**c**) Bar graph showing the proportion of data sets where sex was a significant source of variation by procedure. CSD indicates combined SHIRPA and dysmorphology screen, DEXA: dual-energy X-ray absorptiometry, and PPI: acoustic startle and pre-pulse inhibition. (**d**) Comparison of the consistency of the role of sex in the traits monitored within the DEXA procedure across ten phenotyping centres. BCM: Baylor College of Medicine, HMGU: Helmholtz Zentrum Munich, ICS: Institut Clinique de la Souris, JAX: The Jackson Laboratory, Harwell: Medical Research Council Harwell, NING: Nanjing University, RBRC: RIKEN BioResource Centre, TCP: The Centre for Phenogenomics, UC Davis: University of California, Davis and WTSI: Wellcome Trust Sanger Institute.

environmental factors will contribute to this variability[11]. Previous manuscripts have conducted extensive analysis assessing consistency across institutes and found good agreement in findings[10,12]. It could be argued that the consistency is surprising; for example, considering how critical the microbiome is to phenotypic outcome[13,14]. While these studies are contained within facilities with high biosecurity, the microbiomes will differ from institute to institute. In fact, microbiomes will differ between individual litters depending on the maternal microbiome. This study comparing control data across many litters in effect accounts for this variation, which might go some way to explaining the consistency of the findings across sites.

**Sex as a modifier of a treatment effect.** We next looked at the role of sex in influencing phenotypes in the context of gene ablation (Supplementary Fig. 1c–e). Bespoke statistical analyses, distinct from those implemented on the IMPC portal, were used to assess sexual dimorphism and control the false positive rate. For this analysis, we used data collected from 2,186 mutant mouse lines, first assessing whether genotype significantly influenced phenotype, and if significant whether the effect was

modified by sex. Of the categorical phenotypes that showed a significant genotype effect (0.46% 1,220/266,952 data sets at 5% FDR), 13.3% (162/1,220) were classed as SD at a 20% FDR (Fig. 4a). Our previous investigations[15] found it necessary to use a higher FDR for categorical traits because of the conservative nature of this statistical pipeline and multiple testing burden. For continuous traits, 7.2% (7,929/110,586 at 5% FDR) had a significant genotype effect, of which 17.7% (1,407/7,929 at 5% FDR) were classed as SD (Fig. 4b). Increasing the stringency of the continuous data analysis by decreasing the FDR to 1%, reduced the number of phenotype calls (3.4%; 3,719/110,586 data sets) but we still observed a high proportion (12.0%; 446/3,719) of sexual dimorphism (Fig. 5a). For continuous traits, phenotypes ascertained using mice phenotyped in multiple batches are more robust as data is collected across multiple litters and modelling of environmental variation is more reliable, thereby giving better control of the false positive rate[16]. Focusing only on multi-batch data sets, 8.9% (4,177/46,925) had a significant genotype effect of which 13.8% were classed as SD (Fig. 5b).

The experimental design and the statistical analysis used here were formulated to control the type-one error (false positive) rate, at the expense of sensitivity[15] (Figs 6 and 7). The fact that we

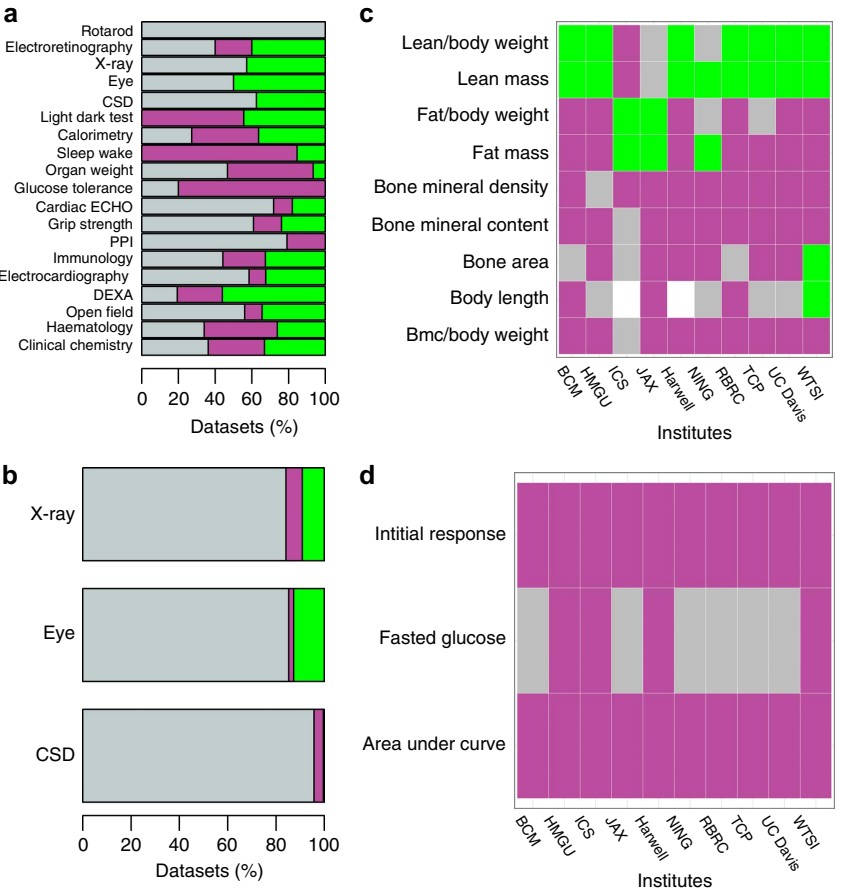

**Figure 3 | Sex as a biological variable after accounting for the potential confounding effect of body weight.** Assessment of the role of sex within an experiment using IMPC wildtype data at a 5% FDR. The analysis assessed the role of sex in the trait of interest, at a centre level. For continuous traits, this was computed as a relative phenotype since weight was included as a covariate. For all sections, green indicates the phenotype was greater in the females, magenta indicates the trait was greater in the males, white indicates missing data and grey indicates there was no significant sex effect. (**a**) Bar graph showing the proportion of data sets where sex was a significant source of variation by procedure for continuous traits. CSD indicates combined SHIRPA and dysmorphology screen, DEXA: dual-energy X-ray absorptiometry and PPI: acoustic startle and pre-pulse inhibition. (**b**) Bar graph showing the role of sex by procedure for categorical traits where CSD indicates combined SHIRPA and dysmorphology screen. (**c**) Comparison of the reproducibility of the sex differences in the traits monitored within the DEXA procedure across ten phenotyping centres. (**d**) Comparison of the consistency of the role of sex in the traits monitored within the intra-peritoneal glucose tolerance test across ten phenotyping centres. BCM: Baylor College of Medicine, HMGU: Helmholtz Zentrum Munich, ICS: Institut Clinique de la Souris, JAX: The Jackson Laboratory, Harwell: Medical Research Council Harwell, NING: Nanjing University, RBRC: RIKEN BioResource Centre, TCP: The Centre for Phenogenomics, UC Davis: University of California, Davis and WTSI: Wellcome Trust Sanger Institute.

detected a significant number of SD genotype–phenotype relationships, despite this limitation and relatively small sample size, suggests that other traits may display more subtle sexual dimorphism. The primary impact of sex as a modifier of genotype effect for continuous traits was that of 'one sex only' (12.8% for all continuous traits using a 5% FDR) where only males or females showed a statistically significant phenotype (Fig. 4b), as demonstrated by the *Usp47*$^{tm1b/tm1b}$ mouse; the mutant that showed the largest proportion of SD calls (Fig. 8 and Supplementary Data 1). Of the SD calls in the IMPC data set, 3.5% demonstrated a phenotype that was significant in both sexes but with opposing phenotypic changes (Fig. 4b); for example a significant increase in the males and a significant decrease in the females (Fig. 8b, total protein and red blood cell). In 0.8% of cases, we observed phenotypes that were significant in both sexes, when compared to controls, but the phenotype was more pronounced in one sex when compared to the other (Fig. 4b). With the goal of assessing the prevalence, a simple summary has been used; however sensitivity will vary by trait. For categorical screens the hit rate by screen averaged 12.5% (s.d. = 3.2%,

Supplementary Table 1), while for continuous data the average SD hit rate by screen was 12.6% (s.d. = 8.3% Supplementary Table 2). Co-correlation of phenotypes is expected and future research will need to focus on cross variable identification of phenotypic abnormalities, but at present is beyond the scope of this manuscript.

Our study focused exclusively on mutants of autosomal loci finding a high proportion associated with one or more SD calls (33.2% of genes studied: 725/2,186). This result is in keeping with the view that once the sex determination cascade is initiated, genes exhibiting SD effects can be located anywhere in the genome[17]. Moreover, it illustrates the pervasive nature of sexual dimorphism that impacts a wide range of loci and genetic systems.

**Sexual dimorphism and gene function.** We considered whether our findings relate to similar examples of SD in humans. Evidence for SD in humans has typically come from complex disease and trait studies where the numbers of tested subjects are amenable to statistical analysis[18]. However, meta-analysis studies have

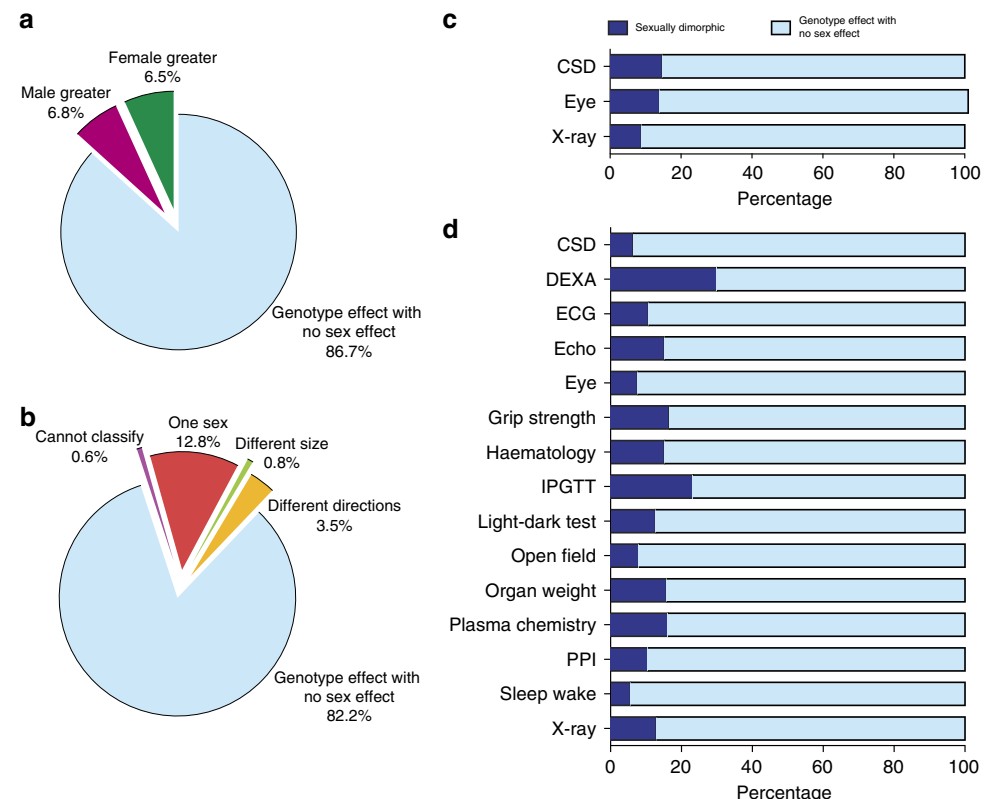

**Figure 4 | Role of sex as a modifier of the genotype effect.** The role of sex in explaining variation in phenotypes of knockout mice as assessed using data from the IMPC. (**a**) Classification for categorical data sets ($n = 1,220$) with a genotype effect at 20% FDR. (**b**) Classification for continuous data sets ($n = 7,929$) with a genotype effect at 5% FDR. The 'Cannot classify' effect arises when statistically an interaction is detected between sex and genotype but the model output is insufficient to specify where the interaction arises. The 'Genotype effect with no sex effect' effect is the classification when the null hypothesis of no genotype*sex interaction was not rejected; this may be because the effect is the same across sexes, or due to a lack of power to detect the differential effect across sexes. (**c,d**) Comparison of SD hit rate for each screen with more than 35 genotype significant hits (**c**) Categorical traits. (**d**) Continuous traits. IPGTT: intra-peritoneal glucose tolerance test; ECG: electrocardiogram; DEXA: dual-energy X-ray absorptiometry; CSD: combined SHIRPA and dysmorphology; PPI: acoustic startle and pre-pulse inhibition.

typically failed to replicate findings with only the association of angiotensin-converting enzyme gene (ACE) and hypertension in men being consistently replicated[18]. Within the IMPC portal, we do relate the knockout phenotypes to Mendelian disease data[19] using resources such as Online Mendelian Inheritance in Man (OMIM)[20] and Orphanet[21] where we are most likely to reproduce the phenotypes observed in these single gene diseases. However, these human resources do not consistently document SD in the signs and symptoms and it is unlikely the numbers of patients recorded for these rare diseases would make detection of significant SD possible.

To determine whether prevalent sex differences are the result of a common biological process, we performed a functional analysis of a set of 29 genes for which sex differences were detected on more than 4% of all measures. The statistical analysis, and subsequent call of SD, is at the level of an individual trait for a genotype. Therefore, classifying a gene as SD is somewhat arbitrary as it involves accounting for the number of traits having a genotypic effect, and the prevalence of SD within these. Despite this limitation, an evaluation of this set of 29 genes in comparison to all curated and experimentally derived functional annotation sets in GeneWeaver[22] revealed statistically significant overlap ($J = 0.0385; P < 1.12 \times 10^{-7}$) to 25 genes associated with 'absence of the oestrous cycle' (MP:0009009), based on representation of *Kiss1r* and *Postn* on both gene lists. A further review of genes for which any significant sex*genotype interaction was detected revealed additional genes associated with MP:0009009, absence of

the oestrous cycle. This additional set includes *Fshr*, *Lhcgr*, *Cyp27b1*, *Fancl* and *Foxo3*. This result suggests that constitutive perturbations of oestrous cyclicity, including developmental absence or loss of cyclicity in adulthood, may broadly influence sex differences. The gene products encoded by *Fshr* (follicle stimulating releasing hormone receptor) and *Lhcgr* (Luteinizing hormone/choriogonadotropin receptor) have well known effects on reproductive cycles, and behaviour due to their role in maintaining hormonal cycles in females. *Cyp27b1* is a steroid synthesizing enzyme, which is primarily involved in vitamin D metabolism, known to influence many sex-specific phenomena in autoimmune and other diseases (for a recent example[23]). *Fancl* (Fanconi anema complementation group L) causes male and female infertility and gonadal hormone abnormalities in Zebrafish[24] through developmental signalling mechanisms via aromatase conversion of androgen. *Foxo3* is associated with ovarian pathology in humans[25] and premature ovarian failure in mice[26]. Therefore, each of these gene perturbations has the capability of influencing hormonal effects on behaviour and physiology, though it remains to be evaluated whether the sex differences herein are stable throughout the hormonal cycle or result from interference in a sex-specific gonadal steroid-regulated process. An evaluation of other genes with high sex difference hit rates may reveal additional pervasive effects on reproductive traits. Other sex differences identified in the IMPC analysis may be the result of more specific effects of gene perturbation on a sex-specific process in males or females.

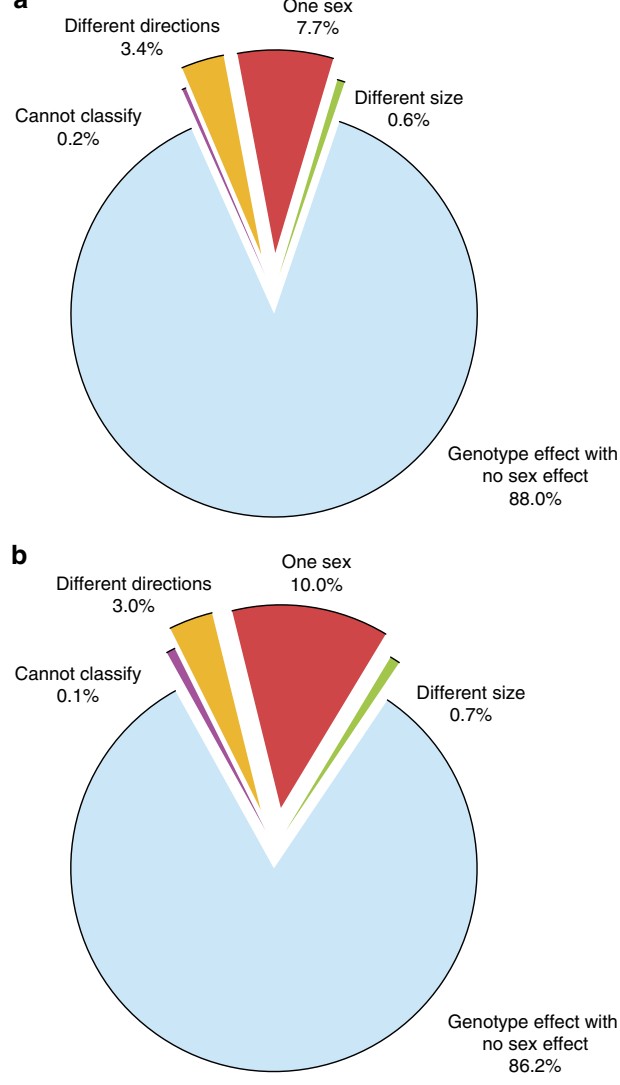

**Figure 5 | Role of sex as a modifier of the genotype effect with more stringent criteria.** Exploration of the role of sex in modifying the genotype effect in studies of continuous traits of knockout mice data from the IMPC. (**a**) Distribution of sex effect in the genotype significant data sets when using a 1% FDR. Overall, 110,586 data sets were tested and 3.4% (3,719) were significant for the stage 1 genotype effect. Of these, 12% (n = 466) were classed as SD. (**b**) Distribution of sex effect in the genotype significant data sets when processing only multi-batch data sets at a 5% FDR. A total of 46,925 data sets were tested and 8.9% (3,719) were significant for the stage 1 genotype effect. Of these, 13.8% (n = 575) were classed as SD.

## Discussion

Many authors have raised the need to address the role of sex in basic biological research and recommendations have been made, but with limited progress to date. Bespoke analysis of IMPC data, revealed that 9.9% of qualitative and 56.6% of quantitative data sets were SD in wildtype mice. Furthermore, as a mediator of a mutant phenotype, sex modifies the genotype effect in 13.3% of qualitative data sets and up to 17.7% of quantitative data sets. Our findings are consistent with a recently published study examining the role of sex in human genetic variation that found the effect was of modest magnitude but across a broad spectrum of traits[27]. Further studies to understand the biological mechanism of the interactions reported herein are challenging because of the difficulty of designing experiments with sufficient sensitivity to consistently detect those interactions. However, our findings also span a broad phenotypic spectrum and indicate that regardless of research field or biological system, consideration of sex is important in the design and analysis of animal studies for studies where sex differences could occur, thus supporting the recent National Institute of Health mandate to consider sex as a biological variable[28].

## Methods

**Methodology consideration.** Bespoke methods were developed to assess for prevalence of sexual dimorphism and are independent of the methodologies implemented on the IMPC portal.

**Ethical approval.** Institutes that breed the mice and collect phenotyping data are guided by their own ethical review panels and licensing and accrediting bodies, reflecting the national legislation under which they operate. Details of their ethical review bodies and licences are provided in Supplementary Table 3. All efforts were made to minimize suffering by considerate housing and husbandry. All phenotyping procedures were examined for potential refinements that were disseminated throughout the consortium. Animal welfare was assessed routinely for all mice involved.

**Mouse generation.** Targeted ES cell clones obtained from the European Conditional Mouse Mutagenesis Program (EUCOMM) and Knockout Mouse Project (KOMP) resource[29,30] were injected into BALB/cAnN, C57BL/6J, CD1 or C57BL/6N blastocysts for chimera generation. The resulting chimeras were mated to C57BL/6N mice, and the progeny were screened to confirm germline transmission. Following the recovery of germline-transmitting progeny, for the majority of lines, heterozygotes were intercrossed to generate homozygous mutants[10]. A few knockout lines were generated on other genetic backgrounds, as detailed in the data output and presented on the IMPC portal. For these lines, control data from the equivalent genetic background was collected. All lines are available from http://www.mousephenotype.org/.

**Genotyping and allele quality control.** The targeted alleles were validated by a combination of short-range PCR, qPCR and non-radioactive Southern blot, as described previously[31,32].

**Housing and husbandry.** Housing and husbandry data was captured for each institute as described in Karp et al.[33] and is available on the IMPC portal (http://www.mousephenotype.org/about-impc/arrive-guidelines).

**Phenotyping data collection.** We have used data collected from high-throughput phenotyping, which is based on a pipeline concept where a mouse is characterized by a series of standardized and validated tests underpinned by standard operating procedures (SOPs). The phenotyping tests chosen cover a variety of disease-related and biological systems, including the metabolic, cardiovascular, bone, neurological and behavioural, sensory and haematological systems and clinical chemistry. The IMPRESS database (https://www.mousephenotype.org/impress), defines all screens, the purpose of the screen, the experimental design, detailed procedural information, the data that is to be collected, age of the mice, significant metadata parameters, and data quality control (QC).

**Experimental design.** At each institute, phenotyping data from both sexes is collected at regular intervals on age-matched wildtype mice of equivalent genetic backgrounds. Cohorts of at least seven homozygote mice of each sex per pipeline were generated. If no homozygotes were obtained from 28 or more offspring of heterozygote intercrosses, the line was classified as non-viable. Similarly, if <13% of the pups resulting from intercrossing were homozygous, the line was classified as being subviable. In such circumstances, heterozygote mice were analysed in the phenotyping pipelines. The random allocation of mice to experimental group (wildtype versus knockout) was driven by Mendelian inheritance. The individual mouse was considered the experimental unit within the studies. Further detailed experimental design information (for example, exact definition of a control animal) for each phenotyping institute, or the blinding strategy implemented is captured with a standardized ontology as detailed in Karp et al.[33] and is available from the IMPC portal (http://www.mousephenotype.org/about-impc/arrive-guidelines).

As a high-throughput project, the target sample size of 14 animals (seven per sex) per knockout strain is relatively low. This number was arrived at after a community-wide debate to find the lowest sample size that would consume the least amount of resources while achieving the goal of detecting phenotypic abnormalities[10]. At times, viability issues or the difficulty in administering a test might further limit the number of animals. As such, whenever data are visualized, the number of animals phenotyped is listed. In a high-throughput environment, replication of individual lines is not cost effective. Instead, we are generating and characterizing a common set of six 'reference' knockout lines that will present a wide range of phenotypes based on previously published research.

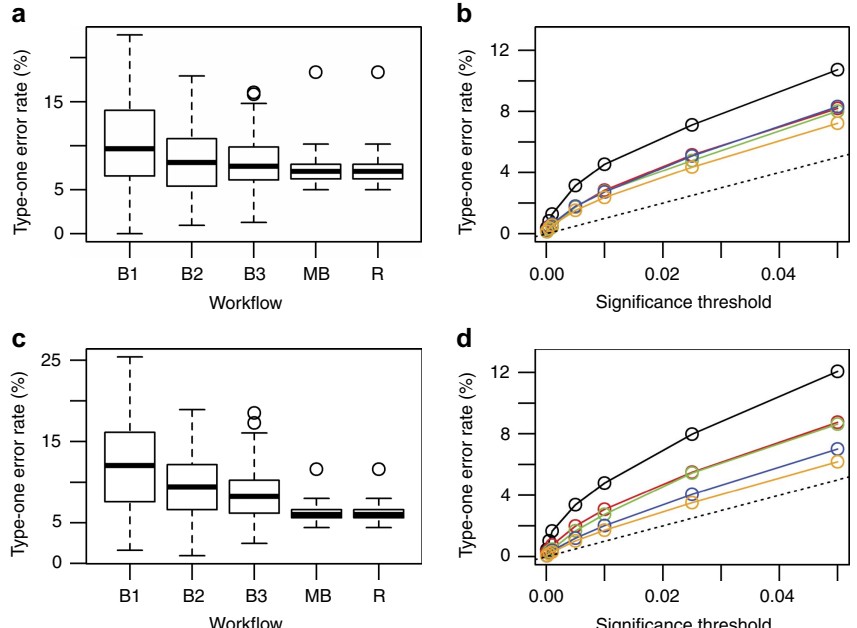

**Figure 6 | Control of type-one errors for continuous traits in the stage 1 assessment of genotype and stage 2 assessment of genotype\*sex interaction.** Resampling studies of wildtype data, to build data sets with 'mock' knockout animals under various phenotyping workflows, were used to assess the control of type-one errors for the statistical pipeline. (**a**) The stage 1 type-one error rate at a trait level at the 0.05 significance threshold as a function of workflow. (**b**) The stage 1 average type-one error rate for various significance thresholds and workflows. (**c**) The stage 2 type-one error rate at a trait level at the 0.05 significance threshold as a function of workflow. (**d**) The stage 2 average type-one error rate at a trait level for various significance thresholds and workflows. For **a**–**d** the one-batch workflow is represented by either B1 or a black line, a two batch workflow is represented by B2 or a red line, a three batch workflow is represented by B3 or a green line, a multiple batch workflow is represented by MB or an orange line, a dashed line represents the ideal rate, and finally a random workflow is represented by R or a blue line.

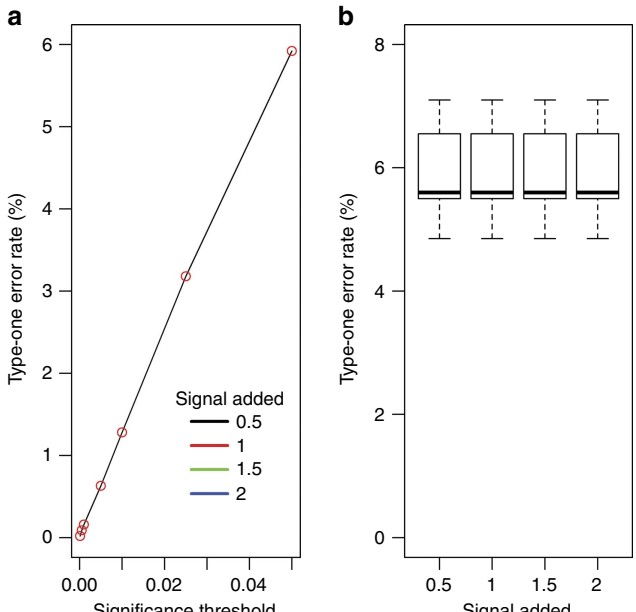

**Figure 7 | Control of type-one errors for assessing a genotype by sex interaction in the presence of a genotype effect for continuous traits.** A simulation study, to assess the stage 2 type-one error rate, where a genotype effect affecting both sexes has been added to the knockout mice as signal (as a function of standard deviation for four levels: 0.5, 1, 1.5, 2). (**a**) The average stage 2 type-one error rate as a function of the significance threshold. Only one level can be seen as the lines overlay each over. (**b**) The stage 2 type-one error rate at the trait level for the 0.05 significance threshold as a function of the signal added. Shown is a boxplot giving a five point summary (minimum, first quantile, mean, third quantile and maximum).

**Data QC.** Pre-set reasons are established for QC failures (for example, insufficient sample) and detailed within IMPRESS to provided standardized options as agreed by area experts as to when data can be discarded. A second QC cycle occurs when data are uploaded from the institutes to the Data Coordination Centre using an internal QC web interface. Data can only be QC failed from the data set if clear technical reasons can be found for a measurement being an outlier. Reasons are provided and this is tracked within the database. QC is an ongoing process; therefore, changes in data composition can occur between different data set versions if an institute later identifies an issue with the data. Analysis within this manuscript used IMPC data set version 4.2, published 8th December 2015.

**Wildtype data sets.** Wildtype data sets were assembled for a trait by selecting wildtype mice that were collected at the same institute, on the same genetic background, the same pipeline and with the same metadata parameters (for example, instrument). The subsequent statistical pipeline required that data was available for both sexes and there were more than 100 data points per sex. The nearest body weight measure was associated with data provided it was within $+/-$ 4 days of the collection of the trait of interest.

**Wildtype-knockout data sets.** Wildtype-knockout data sets were assembled by selecting data from wildtype mice to associate to the data from the knockout mice that were collected at the same institute, from the same genetic background, the same pipeline, and with the same metadata parameters (for example, instrument). The nearest body weight measure was associated with the data provided it was within $+/-$ 4 days of the collection of the trait of interest. A data set was only assembled for a knockout line and trait if data was available on both sexes, there were greater than five readings for each sex for the knockout mice, and body weight data was available. The requirement of a minimum of five readings was to maintain sensitivity.

This process gave 110,586 wildtype-knockout data sets monitoring continuous traits from 10 phenotyping centres. For categorical traits, 266,952 wildtype-knockout data sets from 10 centres were returned. The raw data are available at the IMPC web portal and there is a page detailing the various methods by which data can be extracted from the portal (http://www.mousephenotype.org/data/documentation/index). For Fig. 8 of the manuscript, a gene set was determined by grouping data by phenotyping centre, pipeline, allele, background strain and zygosity. The number of mice that comprise the $Usp47^{tm1b(EUCOMM)Wtsi}$ (MGI:5605792) data set presented within Fig. 8 of the manuscript are shown in Supplementary Table 4.

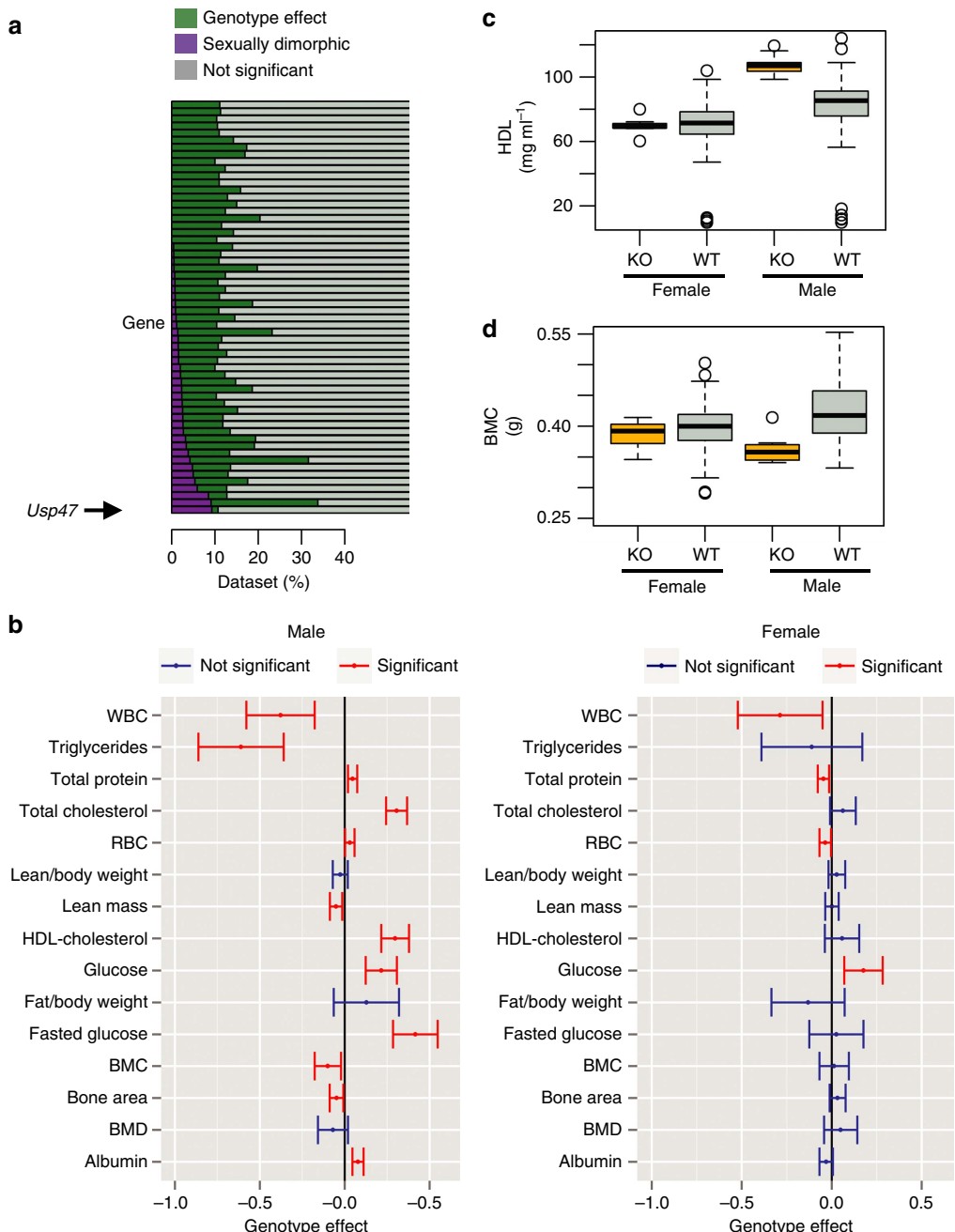

**Figure 8 | Role of sex as a modifier of a genotype–phenotype relationship. (a)** Looking across IMPC knockout lines identified as having 10% or more traits with a significant genotype effect, the proportion of traits classed as SD varied. Data sets are classified as SD (purple) or genotype effect with no sex effect (green) where a genotype effect in a mutant line is observed. **(b)** For all traits identified as having a significant genotype effect for the *Usp47tm1b(EUCOMM)Wtsi* line (MGI:5605792), a comparison is presented of the standardized genotype effect with 95% confidence interval for each sex with no multiple comparisons correction. Standardization, to allow comparison across variables, was achieved by dividing the genotype estimate by the signal seen in the wildtype population. Shown in red are statistically significant estimates. RBC: red blood cells; BMC: bone mineral content; BMD: bone mineral density; WBC: white blood cells. **(c,d)** Raw data visualization for two traits in *Usp47tm1b(EUCOMM)Wtsi* mice identified as having a significant SD genotype effect as the effect was specific to male mice. HDL: high-density lipoprotein–cholesterol; KO: knockout, WT: wildtype. While the *Usp47tm1b(EUCOMM)Wtsi* line had a high proportion of SD traits, the standardized effect size (ES) change leading to a SD call observed for each trait was typically ±1 s.d. unit from the average ES. Globally for all SD calls, the ES was 0.28 (s.d. = 0.38) while for *Usp47tm1b(EUCOMM)Wtsi* the ES was 0.18 (s.d. = 0.14).

**Statistical analysis.** The analysis methods used were developed specifically to answer the biological question of the prevalence of SD and therefore are distinct from the statistical output presented on the IMPC portal. For each statistical analysis a flow diagram summarizing the analysis pipelines is available in Supplementary Fig. 1.

For continuous variables, regression analysis is necessary to assess the effect after accounting for sources of variation such as batch. As such, the estimated effect observed in the regression model cannot always be seen when visualizing raw graphs (Supplementary Fig. 1f).

**Sex as a biological variable for categorical wildtype data.** For categorical traits, the data were recoded to 0 to represent 'as expected' phenotypes or 1 to represent 'not as expected' phenotypes. A Bias Reduction Logistic Regression[34] was used to

assess the impact of sex on the abnormality rate with a likelihood ratio test that compares a test model ($Y \sim$ sex) with a null model ($Y \sim 1$). The $P$ values were adjusted for multiple testing using the Hochberg method to control the FDR to 5%. To assess the biological effect, the differences in two binomial proportions were calculated and the 95% confidence interval calculated utilizing the Newcombe's method. The analyses assessing the role of sex on categorical data, assumes that batch to batch and litter variation is negligible as discussed in ref. 15.

**Sex as a biological variable for continuous wildtype data.** For continuous traits, to assess the role of sex after adjusting for potential body weight differences, a mixed model regression analysis was used with model optimization to select the covariance structure for the residual to the data with a likelihood ratio test that compares a test model (equation 1) with a null model (equation 2). The $P$ values were adjusted for multiple testing using the Hochberg method to control the FDR to 5%.

$$Y \sim \text{Sex} + \text{Weight} + (1|\text{Batch}), \qquad (1)$$

$$Y \sim \text{Weight} + (1|\text{Batch}). \qquad (2)$$

The role of sex was also assessed as an absolute difference using a mixed model regression with a likelihood ratio test to compare a test model ($Y \sim \text{Sex} + (1|\text{Batch})$) with a null model ($Y \sim (1|\text{Batch})$). The analyses assume that batch is a source of variation that adds noise in an independent normally distributed fashion. When weight is included, the analysis assumes that there is a linear relationship between weight and the variation of interest. Analysis was restricted to data sets with more than 100 data points per sex, and thus would be a data set comprising multiple batches and therefore would be robust to the analysis[15].

Using the output from the two pipelines assessing the role of sex as a source of variation, the reproducibility of the role of sex across institutes was assessed for a variable that had been measured at three or more institutes. As data sets have differing size, and sensitivity varied across institutes, discordant results were classed as those where the effect of sex was in opposing direction.

**Sex as a modifier of genotype effect-in categorical data.** For each trait of interest, the data were recoded to either 0 to represent 'as expected' phenotypes or 1 to represent 'not as expected' phenotypes. The statistical pipeline, comparing the abnormality rates in the knockout mice against the baseline population was optimized to maximize sensitivity whilst maintain control of the type-one errors. In summary, a two stage process was used where first the genotype role was assessed and, if statistically significant, then the genotype effect by sex was assessed. To reduce the multiple testing burden, potential filters were used to allow analysis of only data sets that have the potential for statistical significance to be queried. To assess potential for a genotype effect, the Mantel–Haenszel alpha star (the minimal attainable $P$ value for a data set) was calculated. If a data set had potential ($P<0.05$), then the role of genotype was assessed using a one-sided Cochran–Mantel–Haenszel mid $P$ value to compare the proportion of abnormalities events difference between the knockout and wildtype groups, stratified by sex. After multiple testing adjustments, using the Hochberg method to control the FDR to 5%, data sets were selected for stage 2 testing of an interaction. The interaction was assessed by comparing abnormality rates between the sexes of the knockout data only using a Bias Reduction Logistic Regression with a likelihood ratio test that compared a test model ($Y \sim$ sex) with a null model ($Y \sim 1$). Prior to the assessment, the potential was assessed using a LR_KO alpha star $P$ value, defined as the most extreme $P$ value possible arising as a function of the number of abnormal calls and number of readings within a data set was used as a filter to select data sets for statistical testing for stage 2 ($P<0.05$). The remaining $P$ values were adjusted for multiple testing using the Hochberg method to control the FDR to 20%. Lines were selected on statistical significance. To assess the biological effect, the difference in two binomial proportions was calculated and the 95% confidence interval calculated utilizing the Newcombe's method. For data sets with a significant effect at stage 1, but not stage 2, the change was classified as 'genotype effect with no sex effect' as there was evidence of a genotype effect but the genotype*sex interaction was not significant, whilst those which were significant at stage 2 the change was classified ('Female greater' or 'Male greater') by comparing the abnormality rates in the knockout mice by sex. The analyses assessing the impact of genotype ablation on categorical data, assumes that batch to batch and litter variation is negligible as discussed in ref. 15.

The C57BL/6NTac strain carries the $Crb1^{Rd8}$ mutation[35]. This recessive single base pair mutation (Retinal degeneration 8) can lead to a mild form of retinal degeneration that affects vision. The onset of the phenotype appears to be between 2 and 6 weeks of age[36]. The IMPC consortium within the eye screen monitors abnormalities including various retina parameters. The statistical analysis compares the abnormality rate in the knockout to the wildtype within that institute to account for variation in the penetrance of the retinal degeneration in the baseline.

The SD hit rate comparison across screens excluded screens with <35 hits.

**Sex as a modifier of the genotype effect in continuous data.** A two stage pipeline was implemented; stage 1 assessed the role of genotype and stage 2 assessed whether sex interacted with genotype. The complexity of the model is limited by the low number of knockout mice used; as such key fixed effects have been selected and batch is treated as a random effect[16,37]. For stage 1, testing the role of genotype, a mixed model regression analysis was used with model optimization to selects the covariance structure for the residual. The genotype effect was assessed with a likelihood ratio test comparing a full model (equation 3) with a null model (equation 4). The resulting $P$ values were adjusted for multiple testing using the Hochberg method to control the FDR to 5%. For stage 2, testing the role of sex, a mixed model regression analysis was used with model optimization to select a covariance structure for the residual. The interaction was assessed with a likelihood ratio test comparing a full model (equation 3) with a null model (equation 5). The resulting $P$ values were adjusted for multiple testing using the Hochberg method to control the FDR to 5%.

$$Y \sim \text{Genotype} + \text{Sex} + \text{Genotype}*\text{Sex} + \text{Weight} + (1|\text{Batch}), \qquad (3)$$

$$Y \sim \text{Sex} + \text{Weight} + (1|\text{Batch}), \qquad (4)$$

$$Y \sim \text{Genotype} + \text{Sex} + \text{Weight} + (1|\text{Batch}), \qquad (5)$$

$$Y \sim \text{Sex} + \text{Genotype}:\text{Sex} + \text{Weight} + (1|\text{Batch}). \qquad (6)$$

A final model was fitted (equation 6) to estimate the genotype effect by sex which was used to classify the genotype effect. The estimated genotype effect for each sex, and associated standard error, was standardized by dividing the values by the average of the average wildtype male and female mice to allow comparison across traits. Data sets were also given a workflow classification depending on how the knockout data were collected. Multi-batch data sets were defined as those with four or more distinct batches consisting of three or more batches within one sex and two or more for the other sex. One-batch data sets were defined as those with knockout mice collected in one batch. All other workflows were classed as low-batch. When a genotype effect was detected for a data set, the effect was classified as described in Supplementary Fig. 1d. For example, if a data set was significant at stage 1 but not for stage 2, as there was evidence of a genotype effect but the genotype*sex interaction was not significant, the effect would be classified as 'genotype effect with no sex effect'.

The calls were reviewed individually by biologists to validate the calls made by the computational pipeline. Where questions were raised on a computational call, if a statistical issue could be identified (for example, a continuous variable was bound and thus was not appropriate for a mixed model methodology) then all data sets for that variable were removed. See the list detailed in the available code[38].

The analyses assume that batch is a source of variation that adds noise in an independent normally distributed fashion. When weight is included, the analysis assumes that there is a linear relationship between weight and the variation of interest and that the slope doesn't depend on sex. To address the concern that an interaction between weight and sex could act as a confounder, the data was processed with a model with an additional weight*sex term. The results were equivalent to that seen without the inclusion of the term (data not shown, but available at ref. 38). To validate the analysis pipeline, the control of type-one errors was investigated by a series of resampling studies of wildtype data from Wellcome Trust Sanger Institute MouseGP pipeline under the null at both stage 1 and 2 as described in ref. 16. Wildtype data was taken from five procedures (clinical chemistry, dual-energy X-ray absorptiometry (DEXA), immunophenotyping, haematology and open field) giving 60 traits. The simulated wildtype-knockout data sets were then examined statistically to assess the type-one error rate control at stage 1 and 2.

The control of type-one errors for stage 2 was also assessed under the null for stage 2 in the presence of a genotype effect that affected both sexes equally. Simulated data was constructed based on the signal characteristics (mean, variance and sex effect) of five clinical chemistry traits to give 14 male and 14 female data points in 300 batches. Batch variation was simulated under the assumption it was normally distributed with mean zero and defined variance that was 25% of the estimated s.d. Body weight data was generated by random sampling from the average signal for a wildtype female mouse. Resampling studies mimicking a random workflow were run as described in ref. 16 to build wildtype-knockout data sets (iterations 2,000). Signal was added to the knockout mice as a proportion of standard deviation (0, 0.5, 1, 1.5, 2) to represent a main effect genotype effect. The resulting data set were then examined statistically to assess the type-one error rate at stage 2.

The SD hit rate comparison across screens excluded screens with <35 hits.

**Enrichment analysis.** A list of genes (GSID = GS248996) and their SD hit rate (per cent of measures showing a statistically significant sex*genotype interaction) were entered into the GeneWeaver database. Genes with a 4% or greater hit rate (GSIDS = GS248973) were stored in a gene set. A 'search for similar gene sets' was performed using Jaccard similarity of GS248973f against GeneWeaver's database of >100,000 gene sets from multiple sources including gene expression studies, curated annotations and other genomic data resources[22]. A statistically similar gene set was compared to the larger set of all sex*genotype interactions (GS248996) using the Jaccard similarity analysis tool to find additional relevant genes.

**Data availability.** All data sets, scripts and output have been made available at www.mousephenotype.org/data/sexual-dimorphism and as a Zenodo repository at http://doi.org/10.5281/zenodo.260398 (ref. 38).

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

## Acknowledgements

We thank the scientists, technicians, informaticians, programmers and managers across the IMPC who have generated and disseminated this data. We thank Dr David Adams for his assistance and insight with manuscript preparation.

## Author contributions

N.A.K. conceived the question, developed the methodology, analysed the data, validated the results and drafted the manuscript. J.M. developed code allowing the IMPC data to be analysed. Y.B., R.H., S.Y., N.K. and R.F.M. were involved in statistical methodology development. The International Mouse Phenotyping Consortium, J.K.W., A.L.B., T.S., Y.H., S.A.M., C.M., A.M.F., K.L.S., M.H.A., H.F., H.M., J.R.S., S.B., S.G., S.W., K.C.K.L., K.P.S., L.B., L.S., D.W., M.E.D., R.R.-S., S.W., T.F.M., H.W., A.-M.M. and H.P. were involved in collected and disseminating the phenotyping data. C.J.L., A.O.S., A.-M.M., H.M., S.G., L.S., S.B. were involved in validating the results. A.-M.M., K.C.K.L., H.M., M.H.A., M.E.D., A.L.B., J.R.S., K.L.S., K.P.S., H.P., S.A.M., C.M., A.M.F., Y.H. were involved in acquisition of funding. N.A.K., A.-M.M., D.J.A., J.E.M., R.F.M., A.O.S., C.M., A.M.F., M.J.J., H.P., K.C.K.L., S.B., J.K.W. were involved in manuscript preparation.

## Additional information

**Competing interests:** Natasha A. Karp is now an employee of AstraZeneca; however, there are no products in development or marketed products related to this work to declare. The remaining authors declare no competing financial interests.

## International Mouse Phenotyping Consortium

Yuichi Obata[29], Tomohiro Suzuki[29], Masaru Tamura[29], Hideki Kaneda[29], Tamio Furuse[29], Kimio Kobayashi[29], Ikuo Miura[29], Ikuko Yamada[29], Nobuhiko Tanaka[29], Atsushi Yoshiki[29], Shinya Ayabe[29], David A. Clary[30], Heather A. Tolentino[30], Michael A. Schuchbauer[30], Todd Tolentino[30], Joseph Anthony Aprile[30], Sheryl M. Pedroia[30], Lois Kelsey[31], Igor Vukobradovic[31], Zorana Berberovic[31], Celeste Owen[31], Dawei Qu[31], Ruolin Guo[31], Susan Newbigging[31], Lily Morikawa[31], Napoleon Law[31], Xueyuan Shang[31], Patricia Feugas[31], Yanchun Wang[31], Mohammad Eskandarian[31], Yingchun Zhu[31], Lauryl M.J. Nutter[31], Patricia Penton[31], Valerie Laurin[31], Shannon Clarke[31], Qing Lan[31], Khondoker Sohel[31], David Miller[31], Greg Clark[31], Jane Hunter[31], Jorge Cabezas[31], Mohammed Bubshait[31], Tracy Carroll[31], Sandra Tondat[31], Suzanne MacMaster[31], Monica Pereira[31], Marina Gertsenstein[31], Ozge Danisment[31], Elsa Jacob[31], Amie Creighton[31], Gillian Sleep[31], James Clark[7], Lydia Teboul[32], Martin Fray[32], Adam Caulder[32], Jorik Loeffler[32], Gemma Codner[32], James Cleak[32], Sara Johnson[32], Zsombor Szoke-Kovacs[32], Adam Radage[32], Marina Maritati[32], Joffrey Mianne[32], Wendy Gardiner[32], Susan Allen[32], Heather Cater[32], Michelle Stewart[32], Piia Keskivali-Bond[32], Caroline Sinclair[33], Ellen Brown[33], Brendan Doe[33], Hannah Wardle-Jones[33], Evelyn Grau[33], Nicola Griggs[33], Mike Woods[33], Helen Kundi[33], Mark N.D. Griffiths[33], Christian Kipp[33], David G. Melvin[33], Navis P.S. Raj[33], Simon A. Holroyd[33], David J. Gannon[33], Rafael Alcantara[33], Antonella Galli[33], Yvette E. Hooks[33], Catherine L. Tudor[33], Angela L. Green[33], Fiona L. Kussy[33], Elizabeth J. Tuck[33], Emma J. Siragher[33], Simon A. Maguire[33], David T. Lafont[33], Valerie E. Vancollie[33], Selina A. Pearson[33], Amy S. Gates[33], Mark Sanderson[33], Carl Shannon[33], Lauren F.E. Anthony[33], Maksymilian T. Sumowski[33], Robbie S.B. McLaren[33], Agnieszka Swiatkowska[33], Christopher M. Isherwood[33], Emma L. Cambridge[33], Heather M. Wilson[33], Susana S. Caetano[33], Cecilia Icoresi Mazzeo[33], Monika H. Dabrowska[33], Charlotte Lillistone[33], Jeanne Estabel[33], Anna Karin B. Maguire[33], Laura-Anne Roberson[33], Guillaume Pavlovic[34], Marie-Christine Birling[34], Wattenhofer-Donze Marie[34], Sylvie Jacquot[34], Abdel Ayadi[34], Dalila Ali-Hadji[34], Philippe Charles[34], Philippe André[34], Elise Le Marchand[34], Amal El Amri[34], Laurent Vasseur[34], Antonio Aguilar-Pimentel[35], Lore Becker[35], Irina Treise[35], Kristin Moreth[35], Tobias Stoeger[35,36,37], Oana V. Amarie[35,38], Frauke Neff[35,39], Wolfgang Wurst[38,40,41,42], Raffi Bekeredjian[43], Markus Ollert[44,45], Thomas Klopstock[41,42,46], Julia Calzada-Wack[35,39], Susan Marschall[35], Robert Brommage[35], Ralph Steinkamp[35], Christoph Lengger[35], Manuela A. Östereicher[35], Holger Maier[35], Claudia Stoeger[35], Stefanie Leuchtenberger[35], Ali Ö. Yildrim[35,36,37], Lillian Garrett[35,38], Sabine M. Hölter[35,38], Annemarie Zimprich[35,38], Claudia Seisenberger[38], Antje Bürger[38], Jochen Graw[38], Oliver Eickelberg[36,37], Andreas Zimmer[47], Eckhard Wolf[48], Dirk H. Busch[49], Martin Klingenspor[50,51], Carsten Schmidt-Weber[52], Valérie Gailus-Durner[35], Johannes Beckers[35,53,54], Birgit Rathkolb[35,48,54], Jan Rozman[35,54]

[29]BioResource Center, RIKEN, 3-1-1 Koyadai, Tsukuba, Ibaraki 305-0074, Japan; [30]Mouse Biology Program, University of California, 2795 Second Street, Suite 400, Davis, California 95618, USA; [31]The Centre for Phenogenomics, 25 Orde Street, Toronto, Ontario, Canada M5T 3H7; [32]MRC Harwell Institute, Harwell Campus, Harwell OX11 0RD, UK; [33]Mouse Genetics Project, The Wellcome Trust Sanger Institute, Hinxton, Cambridge CB10 1SA, UK; [34]Institut Clinique de la Souris, 1 Rue Laurent Fries, Illkirch 67404, France; [35]German Mouse Clinic, Institute of Experimental Genetics, Helmholtz Zentrum München, Ingolstädter Landstraße 1, Neuherberg 85764, Germany; [36]Comprehensive Pneumology Center, Institute of Lung Biology and Disease, Helmholtz Zentrum München, German Research Center for Environmental Health, Ingolstädter Landstrasse 1, Neuherberg 85764, Germany; [37]Member of the German Center for Lung Research, Helmholtz Zentrum München, German Research Center for Environmental Health, Neuherberg 85764, Germany; [38]Institute of Developmental Genetics, Helmholtz Zentrum München, Ingolstädter Landstraße 1, Neuherberg 85764, Germany; [39]Institute of Pathology, Helmholtz Zentrum München, Ingolstädter Landstraße 1, Neuherberg 85764, Germany; [40]Technische Universität München-Weihenstephan, c/o Helmholtz Zentrum München, Ingolstädter Landstraße 1, Neuherberg 85764, Germany; [41]Deutsches Institut für Neurodegenerative Erkrankungen (DZNE), Site Munich, München 80336, Germany; [42]Munich Cluster for Systems Neurology (SyNergy), Adolf-Butenandt-Institut, Ludwig-Maximilians-Universität München, Schillerstr. 44, München 80336, Germany; [43]Department of Cardiology, University of Heidelberg, Im Neuenheimer Feld 410, Heidelberg 69120, Germany; [44]Department of Infection and Immunity, Luxembourg Institute of Health, 29, Rue Henri Koch, Esch-sur-Alzette 4354, Luxembourg; [45]Department of Dermatology and Allergy Center, Odense Research Center for Anaphylaxis, Odense University Hospital, University of Southern Denmark, Odense C 5000, Denmark; [46]Department of Neurology, Friedrich-Baur-Institute, University Hospital of LMU Munich, Ziemssenstrasse 1, Munich 80336, Germany; [47]Medical Faculty, Institute of Molecular Psychiatry, University of Bonn, Sigmund Freud Street 25, Bonn 53127, Germany; [48]Institute of Molecular Animal Breeding and Biotechnology, Gene Center, Ludwig-Maximilians-University München, Feodor-Lynen Street 25, Munich 81377, Germany; [49]Institute for Medical Microbiology, Immunology

and Hygiene, Technische Universität München, Trogerstrasse 30, Munich 81675, Germany; [50]Technical University Munich, EKFZ—Else Kröner Fresenius Center for Nutritional Medicine, Gregor-Mendel-Street 2, Freising-Weihenstephan 85350, Germany; [51]ZIEL—Institute for Food and Health, Technical University Munich, Gregor-Mendel-Street 2, Freising-Weihenstephan 85350, Germany; [52]Center of Allergy & Environment (ZAUM), Technische Universität München, and Helmholtz Zentrum München, Ingolstädter Landstrasse, Neuherberg 85764, Germany; [53]School of Life Science Weihenstephan, Technische Universität München, Alte Akademie 8, Freising 85354, Germany; [54]German Center for Diabetes Research (DZD), Ingolstädter Landstr. 1, Neuherberg 85764, Germany.

