## [Peer Review File · Nature Communications]

Reviewers' comments:

Reviewer #1 (Remarks to the Author):

The authors provide an interesting analysis of the IMPC database, looking to determine the prevalence of sexually dimorphic (SD) traits and genotype-phenotype relationships. They describe that SD phenotypic variation is common in wild-type animals but occurs in less than 20% of traits analyzed showed SD with a significant genotype effect. They then performed a clustering study to determine whether frequently SD genes shared a common functional classification, finding that the 29 genes studied were associated with loss of the estrous cycle. Overall, they highlighted the fact that SD effects are important to consider in mouse phenotyping analysis and therefore including both sexes is merited.

Overall, this paper mines an important dataset to provide an estimate of the frequency of SD in mouse biology. The study appears well-conducted with reasonable conclusions drawn. The results are interesting though not especially surprising. My main concern is the pertinence of this information to the study of gene function and SDs based on the data they have provided. It is largely a descriptive set of percentages without insight that informs the reader about the biology or potential impact of the findings.

Concerns:

1) The authors elected to perform the analysis from the perspective of the analyzed traits rather than the genes for which they have knockouts. This gives a somewhat skewed perspective since it highlights the number of traits that vary with sex rather than the number of genes for which there are SD traits (though they mention this briefly in the text). Can the authors rationalize this choice? In fact, the most interesting part of the analysis concerned the breakdown of traits in the *Usp47* knockout and the clustering, both of which focus on the effect of the genes in question.

2) Can the authors provide more of a discussion about why these SD traits are important? In other words, what would be missed if the female or male mice were not included for that cohort? Just because a SD trait is present does not mean it is important biologically or particularly relevant to the gene function. Can the authors provide any correlation between SDs they found in mice with SDs found in humans (on a gene basis)? This would help provide more significance to the analysis. I am particularly concerned about translational significance since the biggest hit was in genes that cause loss of reproductive cycles, which affects mice much more than humans (e.g. ovariectomized females quickly become obese whereas human females post-ovarian failure do not).

3) How do the authors determine the sensitivity of their phenotypic measures in terms of likelihood to detect significant differences? This invariably will differ from trait to trait making an analysis that collects them all mathematically a bit over simplistic. Are their particular traits that have a much higher tendency to appear in the dataset? Do they correlate with each other in terms of biological function or method used to measure them? Can we learn something about the quality of modern phenotyping analysis from this dataset?

4) Characterization and clustering of genes by SD traits and traits by SD genes would be interesting and could extend the results toward an understanding of the underlying biology.

5) Do microbiome differences between facilities impact SD traits? I'm surprised at how consistent the findings were across sites despite what I assume are large differences in microbiomes.

Reviewer #2 (Remarks to the Author):

This paper examines the effects of sex differences by analysis of high-throughput phenotypic data from 14,250 wild-type and 40,192 mutant mice (representing 2186 knockout lines) obtained from the International Mouse Phenotyping Consortium. The authors used linear modeling to determine the effects of sex on the phenotype variation in wild-type mice. In the case of the mutant mice, the authors first examined if the genotype affected the measured phenotypes and, second, if the phenotypes were affected by sex. The design of this study is quite novel and the analyses generally rigorous. However, I have some suggestions for improvement.

Major issues:

1) They use every phenotype*center as an independent dataset. That is not valid in my opinion - repeating the test for the same trait tested in a different center is not the same as testing a different phenotype in terms of your statistical expectation of result. It also inflates the number of reported tests and makes it unclear how many actual phenotypes we are talking about each time - 54/545 are these 10 phenotypes significant in every center or 54 phenotypes that each comes out significant in only one center?

I would strongly suggest that they either pool the data using a center as another variable in the model, or, preferably that they test each phenotype only on data from one center (randomly chosen) and use other centers to estimate reproducibility per phenotype, rather than including it as independent dataset.

They also start as distinguishing between dataset and phenotype ("... an individual dataset (a phenotypic test/trait at an individual phenotyping centre)..."), and figure 1, but later on use phenotype instead of dataset "...Bespoke analysis of this dataset revealed that 9.9% of qualitative and 56.6% of quantitative phenotypes were sexually dimorphic in wildtype mice.". This is a very confusing presentation and should be clarified.

2) In general I did not find a table with a clear list of all phenotypes tested, from which center, how many male/female mice per test etc - metadata type of thing. It would be much clearer if they do.

3) For supplementary figure 1, where they describe the pipelines - please add numbers for how many datasets survived each filter.

Minor:

1) "Both sexes equally" should be replaced by no detected sex effect (this is the actual meaning based on figure 1 legend, and current annotation is misleading).

2) For the examination of mutated models (I presume knock outs?) - the proportion of genes that have an effect is surprisingly low (0.5% for categorical, and <15% for continuous) - is this normal?

3) "Our previous investigations (NAK, RH, SY, JKW, YB manuscript in review) found it necessary to use a higher FDR for categorical traits due to the conservative nature of this statistical pipeline and multiple testing burden." - they should describe in short why they change the threshold for this specific case if it is not published.

4) Why is supplementary figure 1 separate from other supplementary figures?

Reviewer #3 (Remarks to the Author):

Review of Karp et al., Nature Comm 2016

It has been evident for some time now that a very large number of sex differences exist on the

genetic level, including in genetically modified mice (for a review showing this fact from neuroscience see Jazin and Cahill, *NN Reviews*, 2010). These sex differences occur in essentially every domain of investigation, and are mostly unanticipated and unexplained, facts which should make them more, not less, important to the field. These sex differences demonstrate that the still-dominant assumption that such sex differences will not exist, therefore that potential sex effects may be safely ignored, is simply not scientifically defensible anymore, at least if we are to argue that our science treats males and females equally.

Getting investigators acknowledge this fact is another matter altogether. To that end, Karp et al appear to be providing a very powerful contribution. They report a meta-analysis of open data from a very large number of wild type and mutant mice. In wild type mice, they have found that 10% of categorical variables were significantly influenced by sex, while a striking 57% of continuous variables were significantly influenced by sex. Sex affected every biological area studied. Furthermore, sex even reversed phenotype in about 9% of the cases. In mutant mice, about 13% of categorical and 18% of continuous showed sex differences. The methodology looks perfectly defensible to me, though I don't claim to be expert on these sorts of statistical analyses.

These results are very powerful, and make the study very important and well worth publishing. Furthermore, as the authors acknowledge, their analysis is very conservative, minimizing risk for Type I errors and therefore necessarily increasing the risk for Type II errors. This means the actual effects of sex are doubtless much more prevalent than even this analysis suggests.

Again, these results are not surprising to those who have been carefully observing the literature the past 15 years or so. The problem is that the vast majority of investigators remain stuck in the 90's or 80's or even 70's on this issue, and have not been paying careful attention. The results of Karp et al might get some to finally wake up.

REVIEWERS' COMMENTS:

Reviewer #1 (Remarks to the Author):

Although I am disappointed that the authors could not provide a more detailed analysis of the SD traits and their biological significance, I accept the authors' explanations about the quantitative limitations of the dataset. However, without this type of analysis, statements like "Our findings show that regardless of research field or biological system, consideration of sex is important in the design and analysis of animal studies." are too strong. For example, there would not be a strong rationale to include male mice in a study of genes that disrupt the estrus cycle (if estrus cycle endpoints are the primary focus). Although this point is self-evident, it highlights the issue with over-generalization about the prevalence of SDs and their importance in phenotyping analyses. The authors should make an effort to temper these conclusions.

Reviewer #2 (Remarks to the Author):

Karpova et al argue in their response to our comment 1 (center*phenotype is not appropriate as independent variable) that

"The vision behind this study was to assess the prevalence of sexual dimorphism within an individual experiment to inform the debate on whether within an individual experiment sex should be considered a biological variable. The goal was not to answer the question "does this variable have a sex effect across institutes", and this would be technically challenging."

Sorry for being a pain - but, I find this to be a crucial point. First, this statement is in direct contradiction to multiple statements throughout the article:

"Here we quantify how often sex influences phenotype by analysing data from 14,250 wildtype animals and 40,192 mutant mice, from 2186 single gene knockout lines, produced by the International Mouse Phenotyping Consortium (IMPC). " (page V lines 20-22 of the revised manuscript).

"Sex as a biological variable" - the first header in the results section and the entire point that they are t

"Regardless of biological area studied, sex was found to have a role (Supplementary Fig.IIc, III3a-b) and whilst the effect of sex was in general reproducible, only 8.7% of variables had opposing effects across the phenotyping centres (Supplementary Fig. III3c-d)." page VI-VII.

So the authors clearly try to assess how often sex is relevant to biological phenotype, not a particular experiment or testing of that phenotype, and this is indeed an important scientific question. Moreover the presented analysis clearly tries to examine reproducibility between centers in the same phenotype (figure II and III). So the basic unit of count should be a phenotype not an experiment, and to me it just seems a different way of aggregating their results, rather than making a more elaborate statistical model for each.

That withstanding, I think this is an important attempt at quantifying sex effect across wide range of phenotypes and centers. I do not have any other comments...

Response to Reviewer #1 concerns:

Overall, this paper mines an important dataset to provide an estimate of the frequency of SD in mouse biology. The study appears well-conducted with reasonable conclusions drawn. The results are interesting though not especially surprising. My main concern is the pertinence of this information to the study of gene function and SDs based on the data they have provided. It is largely a descriptive set of percentages without insight that informs the reader about the biology or potential impact of the findings.

We thank the reviewer for their positive remarks. Our objective in this analysis was to present the prevalence of sexual dimorphism for a typical in vivo study characterizing diverse pathways and biological endpoints. Understanding the prevalence of such effects is critical to inform the debate on in vivo research that is currently occurring; the results are designed to impact in vivo research across disciplines. The scale of the IMPC dataset, with gene deletion as the treatment, provided the ideal dataset to assess the prevalence of sexual dimorphism across diverse biological systems but due to the global nature of the study it was not designed to understand specific gene functions in sexual dimorphism.

Nevertheless, we present several patterns and insights that have been gained, and have incorporated a focused analysis assessing whether the prevalent sex differences are the result of a common biological process. Further gene function analysis is limited for two reasons. Firstly, as discussed within the manuscript, the sensitivity to detect a genotype by sex effect is low with only 7 knockout males and 7 knockout females. Secondly, the statistical analysis, and subsequent call of sexual dimorphism, is at the level of an individual trait for a genotype. Consequently, classifying a gene as generally involved in SD is somewhat arbitrary as it involves accounting for the number of traits having a genotypic effect and the prevalence of SD within these. Therefore, classifying at the gene level has technical challenges that will limit our ability to elucidate gene function. The analysis value comes at an individual gene level, understanding that this effect for this trait is dependent on sex. These technical challenges are discussed within the manuscript.

Concerns:

1) The authors elected to perform the analysis from the perspective of the analyzed traits rather than the genes for which they have knockouts. This gives a somewhat skewed perspective since it highlights the number of traits that vary with sex rather than the number of genes for which there are SD traits (though they mention this briefly in the text). Can the authors rationalize this choice? In fact, the most interesting part of the analysis concerned the breakdown of traits in the *Usp47* knockout and the clustering, both of which focus on the effect of the genes in question.

As discussed above, the analyses at the gene level is technically challenging. This issue is now discussed in the manuscript within the results section to provide a rationalisation of this choice.

2) Can the authors provide more of a discussion about why these SD traits are important? In other words, what would be missed if the female or male mice were not included for that cohort? Just because a SD trait is present does not mean it is important biologically or particularly relevant to the gene function. Can the authors provide any correlation between SDs they found in mice with SDs found in humans (on a gene basis)? This would help provide more significance to the analysis.

We thank the reviewer for these suggestions. We had considered whether our results translated to human disease but reports of SD in human are typically for complex diseases which we would not expect to model with our single-gene knockouts and the Mendelian disease resources do not consistently record whether SD is observed. Acting on the reviewer's concern we have now added a discussion on the challenges we encountered in relating these results to human disease. We have also elaborated further on the known role of some of the genes involved in regulation of the estrous cycle that were identified as being enriched for sexually dimorphic effects.

I am particularly concerned about translational significance since the biggest hit was in genes that cause loss of reproductive cycles, which affects mice much more than humans (e.g. ovariectomized females quickly become obese whereas human females post-ovarian failure do not).

Regarding the concern that the biggest hit was in genes that cause loss of reproductive cycles, one should first note that mammalian phenotype annotation is related to the absence of the reproductive cycle, not necessarily the loss of this cycle in adulthood. Therefore the findings are of greater importance from a perspective of reproductive development and constitutive neuroendocrine effects affecting behaviour, physiology and reproductive traits. Further, in response to the example given by the reviewer, we note that ovariectomy is quite different from ovarian failure (which may be related to other complications in the HPG axis), but it should be noted in the more comparable condition of premenopausal hysterectomy, weight gain is observed in humans female.

3) How do the authors determine the sensitivity of their phenotypic measures in terms of likelihood to detect significant differences? This invariably will differ from trait to trait making an analysis that collects them all mathematically a bit over simplistic.

Yes the reviewer is correct, sensitivity will vary from trait to trait, and in reality from institute to institute. Furthermore, as detailed in the manuscript, the analyses is conservative and the pipeline low in power to detect interaction. The goal of this manuscript is to assess the prevalence despite these limitations to raise the awareness of the role of sex. As such, a simplistic summary method is required.

For sex as a source of variation in the control data we have split the analysis by institute to give an indication of the SD rate seen within an institute (Figure 1c and d) and discussed these results within the manuscript. Furthermore, the prevalence of sex having a role in explaining variation in control data is presented globally (Figure 1), by the screen (e.g. Figure 2), by institute (Figure 1), and the output for each trait for each institute is in the data deposited at Zenodo.

For sex as a modifier of treatment effect, we have added analysis by screen to give a comparison of the SD rate (Figure 4) and discussed these results within the manuscript. We have added Supplementary table 2 to allow readers to explore the SD rate by screen.

Are their particular traits that have a much higher tendency to appear in the dataset? Do they correlate with each other in terms of biological function or method used to measure them?

This is an interesting question. Sex differences were common in our control data for continuous traits and this occurred across screens and therefore biological areas as concluded within the manuscript. The prevalence is high, and that limits the relationship question that could be asked.

We have added analysis by screen to give a comparison of the SD rate (Figure 4) and discussed these results within the manuscript. We have added Supplementary table 2 to allow readers to explore the SD rate by screen. There is correlation within a screen, for example if lean mass is affected, other body composition variables tend to be affected. Co-correlation of phenotypes is expected and future research will need to focus on the cross-variable identification of phenotypic abnormalities but is beyond the scope of this manuscript.

We note that an analysis by trait is challenging with our dataset. This is because across IMPC, some of the screens are institute specific; implementation is captured but can be unique to each centre. Traits can also be difficult to align between centres. Together, this results in traits having a variable depth of

coverage. Furthermore, sensitivity does vary between traits and therefore the difference could be due to a difference in prevalence or sensitivity.

Can we learn something about the quality of modern phenotyping analysis from this dataset?

This is an interesting suggestion. A recent manuscript based on the European consortium phenotypic data (EUMODIC, de Angelis et al., Nature Genetics, 2015) referenced within our manuscript was focused on assessing the quality of high throughput phenotyping. Respectfully, we suggest that additional analysis of phenotypic quality is beyond the scope of this manuscript.

4) Characterization and clustering of genes by SD traits and traits by SD genes would be interesting and could extend the results toward an understanding of the underlying biology.

As discussed above, the analyses at the gene level is technically challenging. This issue is now discussed in the manuscript within the results section to provide a rationalisation of this choice.

5) Do microbiome differences between facilities impact SD traits? I'm surprised at how consistent the findings were across sites despite what I assume are large differences in microbiomes.

We agree with the reviewer that the consistency across sites is surprising. Previous manuscripts (de Angelis et al., Nature Genetics, 2015 and Simon et al., Genome Biology, 2013) have focused extensively on cross centre comparisons and further reflection in this area is beyond the scope of this manuscript. Consistent with these manuscripts, all studies were conducted within facilities with high biosecurity, typically specific-pathogen-free environments, however microbiomes will differ. In fact, microbiomes will differ between individual litters depending on the maternal microbiome. This study comparing control data across many litters in effect accounts for this variation which might go some way to explaining the consistency of the findings across sites. Previous work in inbred mouse strains have shown that genetic differences have far greater influence than facility or sex on the composition of the microbiome. Following the reviewer's query, we have extended the discussion to reflect further on this issue with a particular focus on the microbiome.

Response to Reviewer #2 concerns:

This paper examines the effects of sex differences by analysis of high-throughput phenotypic data from 14,250 wild-type and 40,192 mutant mice (representing 2186 knockout lines) obtained from the International Mouse Phenotyping Consortium. The authors used linear modeling to determine the effects of sex on the phenotype variation in wild-type mice. In the case of the mutant mice, the authors first examined if the genotype affected the measured phenotypes and, second, if the phenotypes were affected by sex. The design of this study is quite novel and the analyses generally rigorous. However, I have some suggestions for improvement.

We thank the reviewer for their positive comments, and we are pleased that they have recognized the novelty of our study and the rigour of our analysis.

Major comments:

1) They use every phenotype*center as an independent dataset. That is not valid in my opinion - repeating the test for the same trait tested in a different center is not the same as testing a different phenotype in terms of your statistical expectation of result. It also inflates the number of reported tests and makes it unclear how many actual phenotypes we are talking about each time - 54/545 are these 10 phenotypes significant in every center or 54 phenotypes that each comes out significant in only one center?

I would strongly suggest that they either pool the data using a center as another variable in the model, or, preferably that they test each phenotype only on data from one center (randomly chosen) and use other centers to estimate reproducibility per phenotype, rather than including it as independent dataset.

The vision behind this study was to assess the prevalence of sexual dimorphism within an individual experiment to inform the debate on whether within an individual experiment sex should be considered a biological variable. The goal was not to answer the question “does this variable have a sex effect across institutes”, and this would be technically challenging. For sex as a source of variation within the control data, some variables can be aligned across centres, but it is limited. The modelling becomes complex as it would be insufficient to only add institute as a parameter; there are metadata parameters unique to each institute that would also require inclusion. To address the concern that the approach inflates the number of reported tests we have generated an additional analysis where we have split the dataset by institute to give an indication of the SD rate seen within an institute and discussed these results within the manuscript.

For sex as a modifier of a genotype effect, most genes were only tested at one centre; consequently a broad analysis across centres is not an option.

They also start as distinguishing between dataset and phenotype (“... an individual dataset (a phenotypic test/trait at an individual phenotyping centre)...”), and figure 1, but later on use phenotype instead of dataset “...Bespoke analysis of this dataset revealed that 9.9% of qualitative and 56.6% of quantitative phenotypes were sexually dimorphic in wildtype mice.”. This is a very confusing presentation and should be clarified.

We agree the wording is a little confusing and has been corrected as suggested.

2) In general I did not find a table with a clear list of all phenotypes tested, from which center, how many male/female mice per test etc - metadata type of thing. It would be much clearer if they do.

In the statistical output in the associated ZENODO repository, detailed in data availability, the number of data points by sex for each analysis and associated metadata (such as institute, genetic background, gene) is captured.

3) For supplementary figure 1, where they describe the pipelines - please add numbers for how many datasets survived each filter.

This is applicable to the categorical analysis pipeline (Supplementary Figure 1C) and has been added to the appropriate figure legend.

Minor comments:

1) “Both sexes equally” should be replaced no detected sex effect (this is the actual meaning based on figure 1 legend, and current annotation is misleading).

Amended to “Genotype effect with no sex effect” where used.

2) For the examination of mutated models (I presume knock outs?) - The proportion of genes that have an effect is surprisingly low (0.5% for categorical, and <15% for continuous) - is this normal?

A Bayesian analyses of the earlier European consortium (EUMODIC) phenotyping data (de Angelis et al., Nature Genetics, 2015) found that the hit rate when clustered by centre and screen (some continuous and some categorical parameters) varied between 0 and 40% when controlling the FDR to 2%. This hit rate is inflated as the analyses did not account for the potential covariate of body weight and the author noted a clustering of phenotypes associated with a body-weight phenotype. Therefore we conclude the hit rate is as expected.

3)"Our previous investigations (NAK, RH, SY, JKW, YB manuscript in review) found it necessary to use a higher FDR for categorical traits due to the conservative nature of this statistical pipeline and multiple testing burden." - They should describe in short why they change the threshold for this specific case if it is not published.

This manuscript is now in print within the journal GENETICS and is referenced directly.

4) Why is supplementary figure 1 separate from other supplementary figures?

Following reformatting, all supplementary figures are now in one document.

Response to Reviewer #3 concerns:

No concerns to address

In response to the remaining reviewer concerns

Reviewer 1: "Although I am disappointed that the authors could not provide a more detailed analysis of the SD traits and their biological significance, I accept the authors' explanations about the quantitative limitations of the dataset. However, without this type of analysis, statements like "Our findings show that regardless of research field or biological system, consideration of sex is important in the design and analysis of animal studies." are too strong. For example, there would not be a strong rationale to include male mice in a study of genes that disrupt the estrus cycle (if estrus cycle endpoints are the primary focus). Although this point is self-evident, it highlights the issue with over-generalization about the prevalence of SDs and their importance in phenotyping analyses. The authors should make an effort to temper these conclusions."

We appreciate the reviewer's concern. We have revised the text in the discussion to clarify that we are only assessing situations where a "sex effect is possible".

Reviewer 2: "Karp et al argue in their response to our comment 1 (center*phenotype is not appropriate as independent variable) that "The vision behind this study was to assess the prevalence of sexual dimorphism within an individual experiment to inform the debate on whether within an individual experiment sex should be considered a biological variable. The goal was not to answer the question "does this variable have a sex effect across institutes", and this would be technically challenging.". Sorry for being a pain - but, I find this to be a crucial point. First, this statement is in direct contradiction to multiple statements throughout the article:

- "Here we quantify how often sex influences phenotype by analysing data from 14,250 wildtype animals and 40,192 mutant mice, from 2186 single gene knockout lines, produced by the International Mouse Phenotyping Consortium (IMPC). "
- "Sex as a biological variable" - the first header in the results section and the entire point that "Regardless of biological area studied, sex was found to have a role (Supplementary Fig.IIc, III3a-b) and whilst the effect of sex was in general reproducible, only 8.7% of variables had opposing effects across the phenotyping centres (Supplementary Fig. III3c-d)."

So the authors clearly try to assess how often sex is relevant to biological phenotype, not a particular experiment or testing of that phenotype, and this is indeed an important scientific question. Moreover the presented analysis clearly tries to examine reproducibility between centers in the same phenotype (figure II and III). So the basic unit of count should be a phenotype not an experiment, and to me it just seems a different way of aggregating their results, rather than making a more elaborate statistical model for each. That withstanding, I think this is

an important attempt at quantifying sex effect across wide range of phenotypes and centers. I do not have any other comments.”

Reviewer 2 had concerns with the aggregation of sexual dimorphic effect at an institute or data type level in the analysis which considered the role of sex within the control data. The concern arose as the wording of the manuscript at times suggests we are assessing the prevalence of sex having a role for a phenotype as an absolute concept rather than within a phenotyping experiment. To address this concern, we have reviewed the manuscript and amended the text to increase clarity. For example:

- **We have clarified the situation more clearly within the introduction by adding that the assessment considered “sex role on a phenotype within a dataset”.**
- **In the first sentence of the second paragraph of the introduction we have amended the text to clarify that we have quantified sex role in influences phenotype within a dataset.**
- **We have amended the title Sex as a biological variable to add the phrase “within an experiment”.**

We believe the manuscript has been much improved following the reviewers input, and we hope that it is deemed suitable for publication in Nature Communications.